# Deep Aggregations of the Polychaete *Amage adspersa* (Grube, 1863) in the Ionian Sea (Central Mediterranean Sea) as Revealed via ROV Observations

**Michela Angiolillo ***,†**, Fabio Bertasi †, Laura Grossi †, Marco Loia, Danilo Vani, Sante Francesco Rende, Michela Giusti and Leonardo Tunesi**

ISPRA—Italian Institute for Environmental Protection and Research, via Vitaliano Brancati 48, 00144 Rome, Italy; fabio.bertasi@isprambiente.it (F.B.); laura.grossi@isprambiente.it (L.G.); marco.loia@isprambiente.it (M.L.); danilo.vani@isprambiente.it (D.V.); francesco.rende@isprambiente.it (S.F.R.); michela.giusti@isprambiente.it (M.G.); leonardo.tunesi@isprambiente.it (L.T.)
* Correspondence: michela.angiolillo@isprambiente.it
† These authors contributed equally to this work.

**Abstract:** Many sessile and tube-dwelling polychaetes can act as ecosystem engineers, influencing the physical–chemical and biological characteristics of their habitats, increasing structural complexity. Thus, they are considered structuring species. In summer of 2021, in southern Sicily (Ionian Sea), benthic assemblages dominated by Ampharetidae *Amage adspersa* were discovered via an ROV survey at a depth range between 166 and 236 m on muddy horizontal seafloor. Large aggregations of this species (up to 297.2 tubes m$^{-2}$), whose tubes are formed from *Posidonia oceanica* debris, occurred alternately with tube-free areas. The area was characterized by the sporadic presence of vulnerable sea pens *Funiculina quadrangularis* (up to 0.08 col. m$^{-2}$) and *Virgularia mirabilis* (up to 0.16 col. m$^{-2}$), and it was possible to detect signs of trawling as well the presence of marine litter (up to 24.0 items 100 m$^{-2}$). The habitat description, distribution, and density of the tubes of *A. adspersa* were assessed via imaging analysis. In addition, morphological diagnostic analyses were carried out on some sampled specimens and on their tubes. The acquired data shed new light on how polychaetes can exploit the dead tissues of *P. oceanica*, contributing to highlight interactions between benthic fauna and seagrass detritus in the marine environment and their ecological role in enhancing the spatial heterogeneity of soft areas of the Mediterranean seafloor.

**Keywords:** *Amage adspersa*; tube building; structuring species; *Posidonia oceanica*; ROV-imaging; marine litter; vulnerable marine ecosystems.

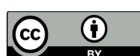

## 1. Introduction

Ecosystem engineering is the formation, alteration, and maintenance of habitats by organisms via the production of physical structures or the transformation of existing materials [1]. In the marine benthic environment, numerous animals function as ecosystem engineers [1,2], constructing intricate frameworks with a high degree of structural complexity that provide refuge to vagile and sessile species [3]. These structuring organisms can profoundly alter the surrounding habitat, affecting larval settlement, regulating community structure by enhancing substrate stability and sediment resuspension, and modulating the current flow velocity and the distribution and availability of resources for other species [4–8]. The most-studied organisms are several filter-feeding taxa such as Cnidarians (i.e., gorgonians, sea fans, bamboo, stony and black corals), massive Poriferans (i.e., sponge grounds), and other colonial invertebrates [9–11]. Nevertheless, organisms with gregarious capacities, such as for example bivalves and

polychaetes, also show the same ability to build complex structures e.g., [8,12–14]. In addition, species that modify sediment have received attention for their direct and indirect effects on the suitability of benthic habitat for other organisms [15,16].

Many polychaetes exhibit a sessile lifestyle and reside within tubes. They often gather in groups and possess the remarkable ability to construct primary biogenic structures or modify the abiotic characteristics of their surrounding substrate [17]. They can provide new substrates for other benthic species, influencing physical, chemical, and biological habitat conditions and regulating ecosystem functioning [14]. Thus, they act as ecosystem engineers, increasing the spatial complexity of hard and soft substrates, and thus are considered structuring species [14]. Dense aggregations of tube-building polychaetes may also represent a favorable food supply for epibenthic predators [18]. On soft areas of the seafloor, the terebellid *Lanice conchilega* can create reef-like structures by cementing sand grains together [19–21]. Many other tube-building macrofaunal species such as *Owenia fusiformis* [18], *Clymenella torquata* [22], *Polydora cornuta* [23], and *Pygospio elegans* [24] have been shown to form dense aggregations reaching densities of up to 200,000 individuals m$^{-2}$. The tube builders *Hobsonia florida* (Polychaeta, Ampharetidae) and *Pseudopolydora kempi japonica* (Polychaeta, Spionidae) facilitate the recruitment of other taxa into 10 cm$^{-2}$ azoic patches [25]. These patches have been shown to be ecologically important, supporting different communities in surrounding areas [22]. The worms, thanks to turbulence caused around their tubes, act as traps for fine particles and larvae [26,27]. Data on polychaetes aggregations are mainly concentrated on shallow-water reefs [17] or on dense patches of tube-building species in sublittoral areas [24]. Some data regarding Sedentaria polychaetes, such as Terebellidae, Oweniidae, Chaetopteridae, Onuphidae, Sabellidae, and Serpulidae being able to structure three-dimensional space on soft seafloor have been provided by [14].

In the marine benthic assemblages, Ampharetidae are among the most common tubiculous polychaetes and play an important role in the ecosystem as surface deposit feeders [28,29]. Many of these polychaetes reside within non-permanent mucous tubes. The members of this family are classified as discretely mobile [29] and can also move to a different location when the surrounding sediment is exploited, either by leaving the old tube and re-burrowing, or simply by lengthening the tube [30]. Many ampharetids have a wide geographical and bathymetric distribution, probably because they can easily adapt to new environments [30]. Although most species are usually found in low quantities [31] in shallower waters, certain species can reach high population densities, forming extensive tube mats e.g., [32,33]. In the deep sea, ampharetids are sometimes among the most abundant species and play a significant role as deep-burrowing organisms, penetrating several centimeters into the sediment [30,34,35]. Among the ampharetids, *Amage adspersa* (Grube, 1863) is a common species in marine sediments, most frequently found between depths of 5 and 114 m [36]. This species, originally described as being from the Adriatic Sea, is common in the Mediterranean Basin [37–39] and has also been found in the northeastern Atlantic Ocean from Iceland [40] and Scotland [41] to Madeira [42] and Senegal [43]. Unlike other species in the family, the tube of *A. adspersa* from Mediterranean sites is distinctively covered with fibrous fragments of *Posidonia oceanica* (L.) Delile, 1813 [38,44]. Dead organic matter from phanerogams, such as *P. oceanica*, is a key resource in marine environments [45]. In sublittoral habitats, dead leaves from *P. oceanica* can sustain a rich fauna of detritivorous crustaceans, mainly amphipods [46] as well as sea urchins [47,48]. In the deep sea, seagrass debris is also utilized as convenient substrate by epifaunal organisms, while its hollow rhizomes offer excellent shelter for various kinds of animals in addition to serving as a source of food for most of them [45].

Within the Marine Strategy Framework Directive (MSFD, 2008/56/EC), monitoring programs are being carried out in Italy to assess the health status and trends of marine ecosystems for the protection of biodiversity [49,50]. A recent exploration carried out with a remotely operated vehicle (ROV) in the mesophotic and deeper waters of the Ionian Sea within the MSFD allowed us to observe dense aggregations of Ampharetidae *A. adspersa*,

providing the opportunity to better characterize a species that is able to form dense tube patches on muddy seafloor.

Over the last two decades, ROV (or other underwater vehicles and towed systems) exploration has proven to be a powerful method for investigating marine ecosystems, providing valuable insights into biodiversity, species distribution, habitat structure, and ecological dynamics i.e., [3,11,51–53]. These technologies enable researchers to explore previously inaccessible or understudied habitats, unveiling new discoveries and expanding our understanding of marine ecosystems, primarily in the deep sea, allowing us to assess ecosystem health, identify vulnerable or threatened species, and evaluate the impacts of human activities.

The present study aims (i) to describe some unreported morphological characteristics of *A. adspersa*; (ii) to characterize the spatial extension and the density patterns of *A. adspersa* aggregations in the study area; (iii) and to describe the megabenthic fauna present in the area. Moreover, the main anthropogenic threats affecting the study area are also discussed.

## 2. Materials and Methods

### 2.1. Study Area

The study area is in the southeastern part of the Sicilian continental margin (Ionian Sea, Central Mediterranean Sea) and intersects with the perimeter of the Italian Plemmirio Marine Protected Area (Figure 1). The insular shelf is narrow, up to 2 to 3 km wide, with its shelf break characterized by the presence of canyon heads. Three main water masses characterize the oceanography of the area [54–56]: In the upper layer (200 m), Modified Atlantic Water (MAW) enters from the Strait of Sicily and flows eastward; in the intermediate layer (~200–700 m), Levantine Intermediate Water (LIW) flows westward toward the Strait of Sicily; in the deepest layer (>700 m) the water of Adriatic origin flows. At the sub-basin scale, the Ionian Sea is also characterized by an inversion of the circulation from cyclonic to anticyclonic (on a decadal scale), referred to as the Adriatic–Ionian Bimodal Oscillating System (BiOS, [57,58]), which influences the surface transport of larvae and litter [55,56]. The area is also characterized by an important and historical fishery tradition of longlines and trawling.

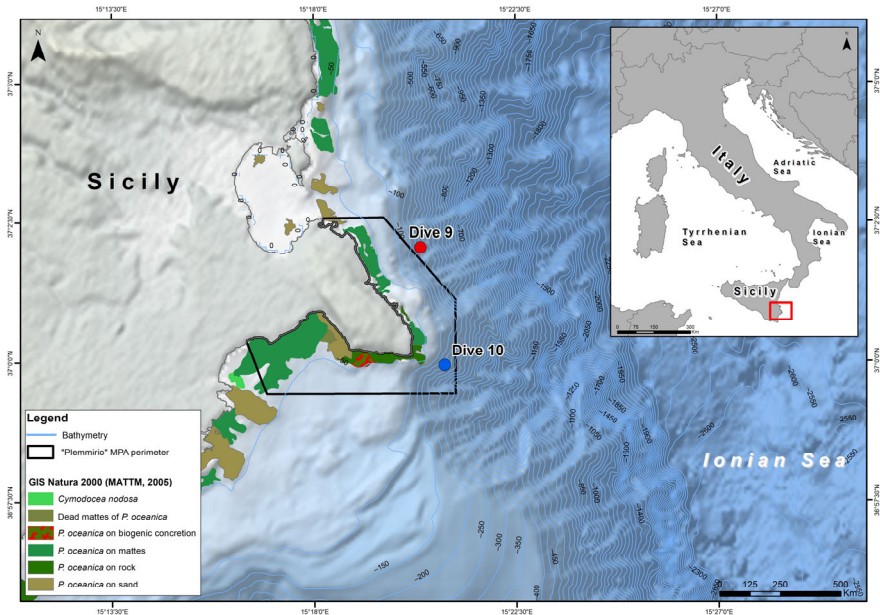

**Figure 1.** Location of study area with main habitat distribution (GIS Natura 2000 [59]) indicated. Black line indicates the perimeter of Plemmirio Italian Marine Protected Area. Background bathymetry was obtained from EMODnet.

### 2.2. ROV Survey

In July 2021, a research campaign was carried out onboard the *R/V Astrea* of *ISPRA*—Italian Institute for Environmental Protection and Research—in the Ionian Sea. During two exploratory dives, carried out in the depth range of 166 to 236 m using a ROV (Perseo, L3 Calzoni), the presence of rich assemblages of *Amage adspersa* tubes was revealed (Figure 1 and Table 1). The ROV was equipped with a Kongsberg high-definition video camera (1920 × 1080 pixels), two lasers 16 cm apart for use as a metric scale, and a manipulator arm used to take biological samples. ROV position was acquired every 2 s using an ultrashort baseline (USBL) underwater compact positioning system (μPAP 200, Kongsberg) with up to 0.25 accuracy. Specific attention was paid to maintaining a constant ROV cruising speed of approximately 0.5 knots and an altitude of approximately 1.5 m from the bottom.

**Table 1.** Summary of remotely operated vehicle (ROV) tracks performed in the study area with an indication of time, geographical coordinates, depth range, total length and total number of sampling units (SUs), *Amage adspersa* tube number (no.), number of SUs with presence of tubes, and tube density (tubes m$^{-2}$), along with litter number (no.) and density (items 100 m$^{-2}$).

|  | **D9** | **D10** |
|---|---|---|
| **Date** | 9 July 2021 | 9 July 2021 |
| **Total time** | 1:00 | 1:04 |
| **Latitude start** | 37°01′47.49″ N | 36°59′54.69″ N |
| **Longitude start** | 15°20′35.14″ E | 15°20′55.14″ E |
| **Depth range (m)** | 166–236 | 120–196 |
| **Track length (m)** | 373 | 651 |
| **SU (no.)** | 15 | 26 |
| **Presence of *Amage* tubes (no. of SU)** | 15 | 15 |
| ***Amage* tube (no.)** | 1373 | 204 |
| ***Amage* tube density (tubes m$^{-2}$ ± se)** | 28.3 ± 9.7 | 2.5 ± 0.7 |
| **Litter items (no.)** | 26 | 23 |
| **Litter density (items 100 m$^{-2}$ ± se)** | 4.78 ± 1.9 | 3.53 ± 1.2 |

### 2.3. Data Analysis

The smooth plot of the georeferenced ROV tracks was imported into the geographic information system (GIS) software program ArcGIS v10.3.1, and each ROV video track was divided into 25 m$^2$ sampling units (SUs; 25 m long and 1 m wide). Overall, 2 h 04 min of ROV footage from the seafloor was processed using the free Internet software program VLC (VideoLAN organization). The portions of the video not relevant (i.e., ascent and descent ROV phases, sample collection, recording close-up images, and frames with poor visibility or that were out of focus) were not considered in the analyses. The substratum type along the tracks was visually classified, and all megafaunal organisms visible in the footage were classified to the lowest taxonomic level, estimating their number in each SU and frequency of occurrence (%). Aggregations of *A. adspersa* tubes were estimated both by occurrence (%) and by density (no. of tubes m$^{-2}$ ± se). Density was calculated in five random frames obtained for each SU, for a total of 150 analyzed frames in D9 (no. = 75 frames) and D10 (no. = 75 frames), respectively (Table 1). ROV laser beams were used as metric scales, and the exact number of specimens was counted using the open-source ImageJ software program (U. S. Government or the National Institutes of Health). For some relevant habitat-forming species, such as the tall sea pen *Funiculina quadrangularis* (Pallas, 1766) and the sea pen *Virgularia mirabilis* (Müller, 1776), the abundance (no. of col. m$^{-2}$) was also calculated in each SU.

The presence of seafloor litter and abandoned, lost, or otherwise discarded fishing gear (ALDFG), along with their interaction with benthic species, was also recorded.

Marine litter items were classified according to the MSFD Joint List [60] into level 1—materials (artificial polymer, cloth/textile, food waste, glass/ceramics, metal, etc.), level 2—use, and level 3—general type. The presence of litter in the footage was evaluated, estimating its quantity (no.), frequency of occurrence (%), and abundance (no. of items 100 $m^{-2} \pm$ se) in all SUs. The litter position and arrangement and the interaction with benthic organisms were classified according to [61].

*2.4. Taxonomic Analysis*

Two tubes containing living specimens were collected from the seabed by the manipulator arm of the ROV. Sampled specimens were immediately placed in 70% ethanol aqueous solution for subsequent morphological analysis.

In the laboratory, worms were gently extracted from the tubes and examined for taxonomic diagnostic characters under a Leica S8-APO stereoscope 50× and a Leica DM2500 compound microscope 1000×. Shirlastain A and methyl green diluted in alcohol 70% was used to stain specimens and enhance the contrast of the external morphological features. Photographs of relevant morphological features were taken to illustrate the descriptions. All the analyses were performed at ISPRA laboratories (Castel Romano, Rome).

Holotype + additional specimen in tube (catalogue no. 5741) was borrowed from the collections of Museum für Naturkunde (MFN), Berlin for comparison with the sampled specimens.

**3. Results**

*3.1. Taxonomy*

Class POLYCHAETA Grube, 1850
Order TEREBELLIDA *sensu* Rouse & Fauchald, 1997
Family AMPHARETIDAE Malmgren, 1866
Amage Malmgren, 1866
*Amage adspersa* (Grube, 1863)

3.1.1. Material Examined

ISPRA.40 (dive D9_ind1), 1 spec. in tube built with *Posidonia oceanica* fibers, length 45 mm, 9 July 2021, 37°01′55.24″ N, 15°20′18.41″ E, depth 214 m; ISPRA.41 (dive D9_ind2), 1 spec. in tube built with *P. oceanica* fibers, length 48 mm, 9 July 2021, 37°01′55.24″ N, 15°20′18.41″ E, depth 214 m.

3.1.2. Comparative Material Examined

Holotype (Figure S1) + additional specimen in tube built with *P. oceanica* fibers (catalogue nr. 5741, MFN)

3.1.3. Description

Length 45–48 mm; width (excl. parapodia) 5–6 mm. Seventeen thoracic chetigers (TC) from segment III, 14 thoracic uncinigers (TU) from segment VI, 13 abdominal uncinigers. Trilobed prostomium with two antero−lateral horns (reported by [38] as "2 *carènes glandulaires divergentes*"). Two eye spots. Buccal tentacles long, smooth, slightly swollen at the tip. All four pairs of pointed branchiae in a single row on segment III; left and right group separated by a small gap. The two lateralmost pairs originate on segment III (TC 1), one pair on segment IV (TC 2), and the innermost pair originate from segment V (TC 3). Branchial length extending backward to TC 12. Notopodia as cylindrical dorsal branch with short cirrus. Ventral shields conspicuous to TU 12, faint to TU 14. Anterior thoracic tori long, gradually shortening towards posterior end, first thoracic torus about three times as long as posterior-most. Abdominal pinnules with small dorsal cirrus. Two long

pygidial cirri with swollen base. Capillary chaetae in two rows, narrow limbate with a fine serration. Thoracic and abdominal hooks all with 3–4 teeth arranged in a single row.

### 3.1.4. Tubes

The tubes are rather long (8–13 cm long, 1.1–1.5 cm wide), cylindrical, and formed from an internal membrane, thin, not very resistant. Its entire external surface is covered with brownish fibers of *P. oceanica* so that the whole tube has a hairy appearance (Figure 2A,B). The section of the tube inserted into the sediment is devoid of vegetal fibers, consisting only of the mucous component (Figure S2), while in the distal part, the fibers are arranged transversely to the tube, densely intertwined to form a smooth and compact inner tube wall (Figure S3).

Many small tubes of other organisms and foraminifera adhering to the fibers (Figures S4 and S5).

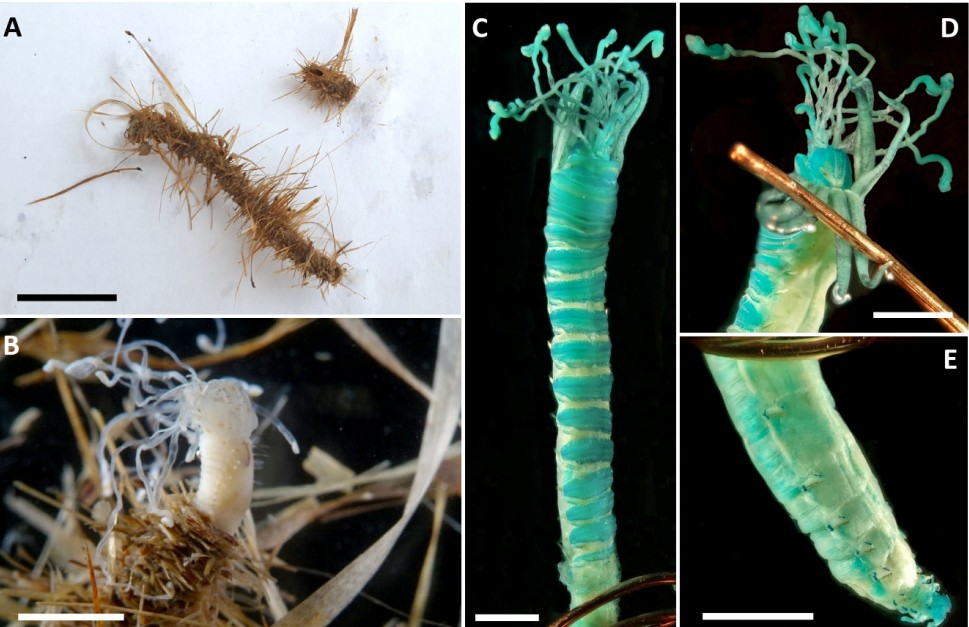

**Figure 2.** Sample of *Amage adspersa* (Grube, 1863) collected in the study area (dive D9_ind2). (**A**) Single tube made of *Posidonia oceanica* fibers; (**B**) *Amage adspersa* with buccal tentacles emerging from the tube. Methyl green staining: (**C**) ventral view of thoracic segments; (**D**) dorsal view of prostomium; (**E**) ventro-lateral view of pygidium. Scale bars: (**A**) = 4 cm; (**B**) = 1 cm; (**C–E**) = 0.5 cm.

### 3.1.5. Methyl Green

Staining homogeneously displaced on peristomium except for the colorless eyespot area. Dorsally, unstained throughout. Ventrally, densely stained on the first 5 segments, then on anterior portion of each segment only up to TC 14–15, the remaining thoracic and abdominal segments unstained. Abdominal short cirrus above pinnules stained dark blue (Figure 2C,D). Pygidium faint green (Figure 2E).

### 3.1.6. Distribution

Species originally described from the Adriatic Sea Lussin-Piccolo (Croatia) [62]. It has been found in other parts of the Mediterranean Sea [37–39] and in the Atlantic Ocean from Iceland [40] and Scotland [41] to Madeira [42] and Senegal [43].

### 3.1.7. Remarks

Length of branchiae in the specimens from Sicily (up to TC 12) is greater than the length in the holotype (up to TC 6–8); at any rate, branchiae were wrinkled in the holotype, probably because of contraction, while in specimens from Sicily, they appeared smooth and elongated. Therefore, we consider this difference in length an artifact of fixation. All the remaining characters appear in the variability range of the species.

### *3.2. ROV Analysis*

#### 3.2.1. *Amage adspersa Aggregations*

The explored sites were mainly characterized by soft seafloor made of muddy sediment (Figure 3). Turbidity was high, and some dead leaves of *P. oceanica* were observed. Large aggregations of *A. adspersa* tubes occurred alternately with tube-free areas on horizontal muddy seafloor. This resulted in the observed structures being highly fragmented, forming large patches or sparse tubes (Figure 3). Minimum and maximum depths ranged from 168 to 235 m, with the densest patches being placed between 195 and 214 m (Figure 4). The majority of the *Amage* tubes were reported from D9 (Figure 4), while the remaining ones occurred with very low frequency in the dive D10 (Table 2 and Figure 4). A total of 1577 specimens were counted, and tube density ranged from 0.0 to 297.2 items m$^{-2}$, with mean values of 28.3 ± 9.7 items m$^{-2}$ and 2.5 ± 0.7 items m$^{-2}$, respectively in D9 and D10 (Table 1). The densest patches presented an extension of about 20 to 30 cm (Figure 3C). Overall, the total investigated surface occupied by *Amage* tubes was estimated to be 184.3 m$^2$, representing up to 23.7% of the total soft seafloor analyzed frames in the Ionian Sea.

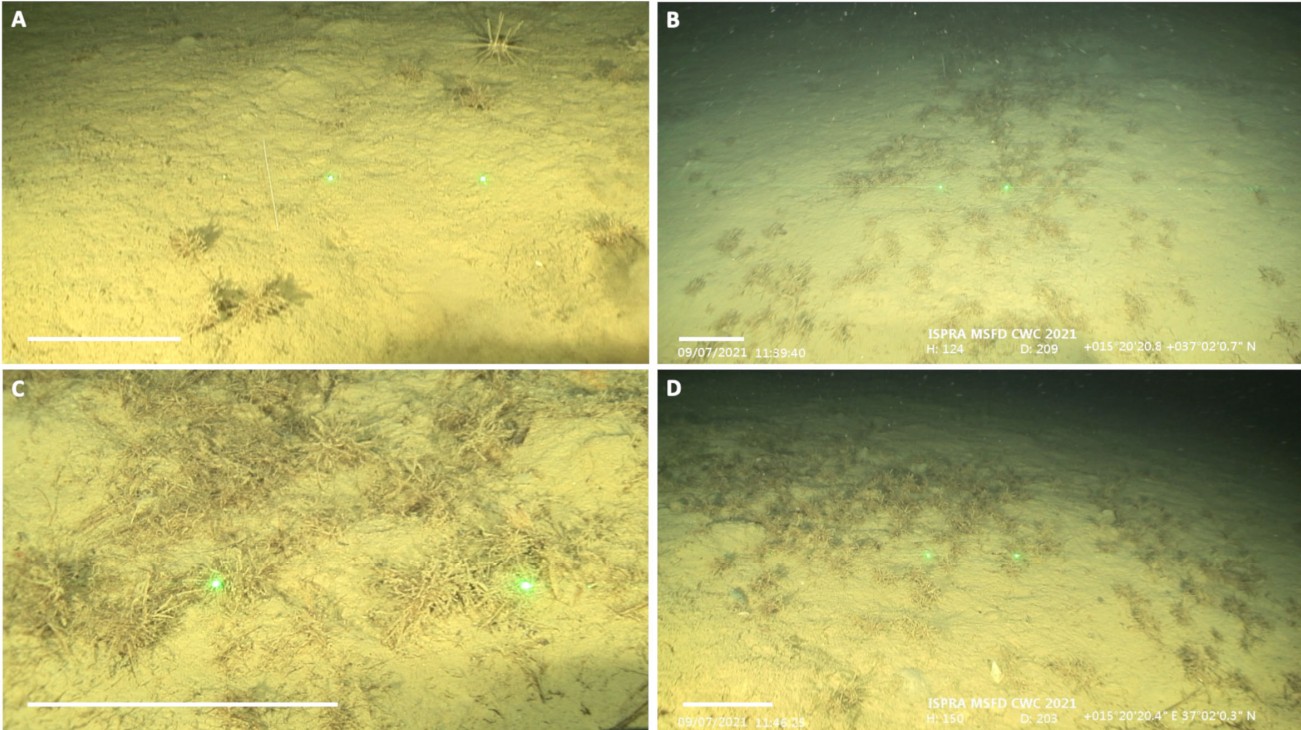

**Figure 3.** Tubes of *Amage adspersa* on muddy seafloor. (**A**) Low density of tubes; (**B**) Dense aggregation of tubes; (**C**) Close-up of a patch; (**D**) Densest patches of tubes. Scale bar: 16 cm.

**Table 2.** List of species and litter items recorded in each dive, with an indication of number of specimens or items (no.) and frequency of occurrence (%) in the SUs.

|  | Taxa | D9 | | D10 | |
|---|---|---|---|---|---|
|  |  | **no.** | **%** | **no.** | **%** |
| Porifera | Porifera ind. | 2 | - | - | - |
| Cnidaria | *Cerianthus membranaceus* (Gmelin, 1791) | 5 | 20 | 33 | 69.2 |
|  | *Funiculina quadrangularis* (Pallas, 1766) | 9 | 33 | 10 | 38.5 |
|  | *Virgularia mirabilis* (Müller, 1776) | 11 | 47 | 8 | 23.1 |
| Annelida | *Amage adspersa* (Grube, 1863) | >1373 | 100 | 204 | 57.7 |
|  | *Bonellia viridis* Rolando, 1822 | 1 | 6.7 | 10 | 11.5 |
|  | Polychaeta ind. | 14 | 33 | 18 |  |
|  | *Myxicola* sp. | 1 | 6.7 | 1 | 3.8 |
| Mollusca | *Sepia officinalis* Linnaeus, 1758 | - | - | 1 | 3.8 |
|  | *Tethys fimbria* Linnaeus, 1767 | - | - | 1 | 3.8 |
|  | *Tonna galea* (Linnaeus, 1758) | - | - | 1 | 3.8 |
|  | *Plesionika* sp. | 14 | 47 | - | - |
|  | *Loligo* sp. | 1 | 6.7 | - | - |
| Echinodermata | Cidaridae | 12 | 53 | 26 | 38.5 |
|  | *Holoturia* sp. | - | - | 1 | 3.8 |
|  | *Parastichopus regalis* (Cuvier, 1817) | 1 | 6.7 | 1 | 7.7 |
|  | Ophiuroidea | 1 | 6.7 | 1 | 3.8 |
|  | *Leptometra phalangium* (Müller, 1841) | 1 | 6.7 | - |  |
| Chordata | *Chlorophthalmus agassizi* Bonaparte, 1840 | 14 | 47 | 10 | 26.9 |
|  | *Helicolenus dactylopterus* (Delaroche, 1809) | 2 | 13 | 5 | 19.2 |
|  | *Lepidorhombus whiffiagonis* (Walbaum, 1792) | 5 | 33 | 2 | 7.7 |
|  | *Lepidorhombus boscii* (Risso, 1810) | - | - | 1 | 3.8 |
|  | *Macroramphosus scolopax* (Linnaeus, 1758) | 4 | 6.7 | 38 | 46.2 |
|  | *Ophisurus serpens* (Linnaeus, 1758) | 1 | 6.7 | 1 | 3.8 |
|  | *Peristedion cataphractum* (Linnaeus, 1758) | 1 | 6.7 | 2 | 7.7 |
|  | *Scorpaena elongata* (Cadenat, 1943) | 1 | 6.7 | 1 | 3.8 |
|  | *Serranus cabrilla* (Linnaeus, 1758) | - | - | 2 | 3.8 |
| Litter | Plastic | 22 | 66.6 | 10 | 23.1 |
|  | Glass/Ceramics | - | - | 9 | 23.1 |
|  | Metal | - | - | 4 | 15.3 |
|  | Ind. | 4 | 13.3 | - | - |

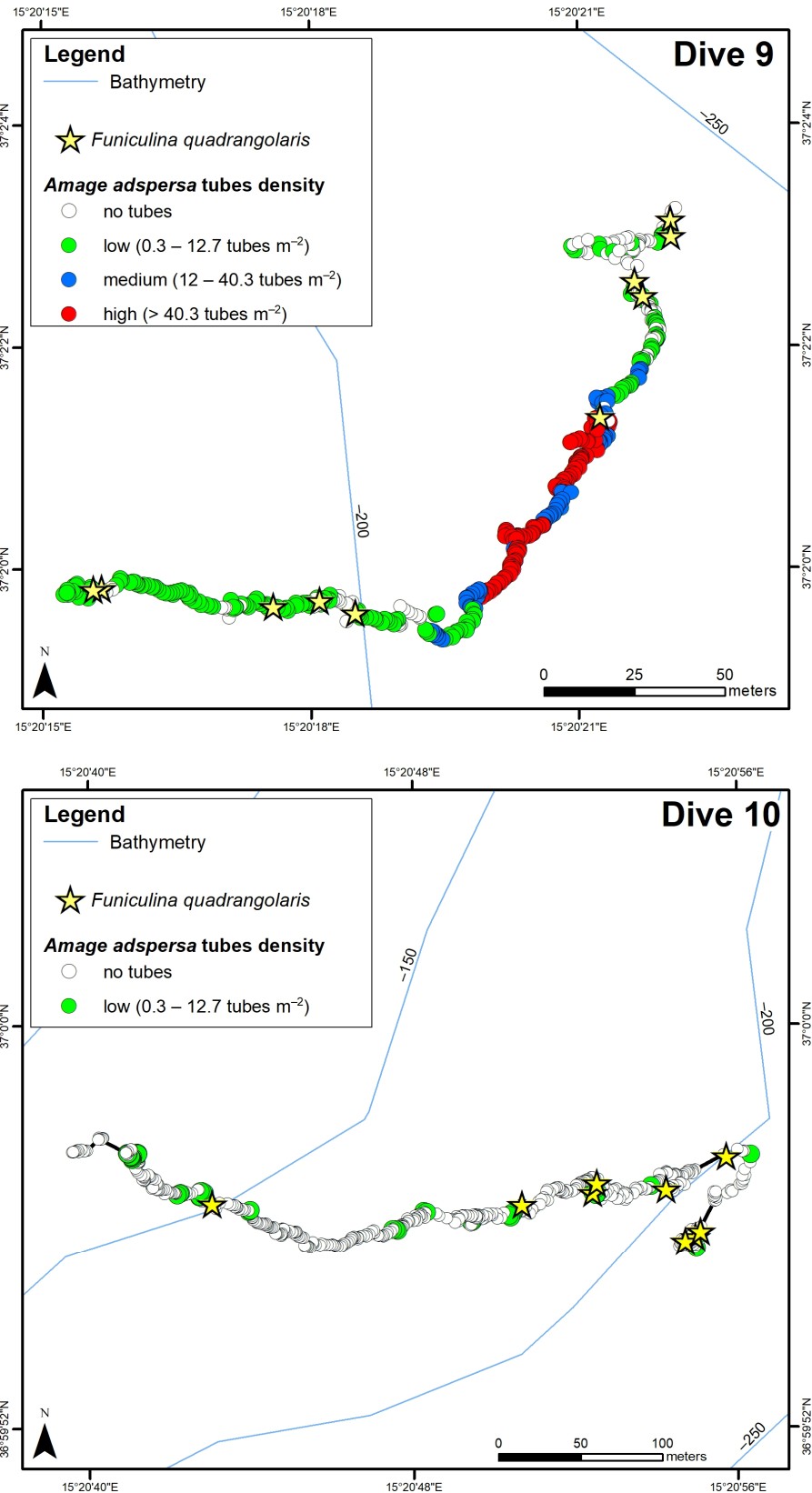

**Figure 4.** Spatial distribution of *Amage adspersa* along the ROV tracks (D9 and D10) with indication of relative species abundance and presence of *Funiculina quadrangularis*.

### 3.2.2. Megafauna

Overall, up to 27 megabenthic species were identified (Table 2), mainly belonging to Porifera (3.7%), Cnidaria (11.0%), Mollusca (19.0%), Annelida (15.0%), Echinodermata (19.0%), and Chordata (33.3%). Most of these species were typical of muddy seafloor habitats and occurred with few specimens at the margin of dense *A. adspersa* aggregations. Relevant vulnerable cnidarian species were observed, namely *Funiculina quadrangularis* (Figure 4), *Virgularia mirabilis,* and *Cerianthus membranaceus* (Gmelin, 1791) (Figure 5A–C). A total of 38 colonies of the first two species were observed (Table 2). The mean density was 0.024 col. m$^{-2}$ ± 0.009 and 0.015 col. m$^{-2}$ ± 0.004 for *F. quadrangularis* (range 0–0.08 col. m$^{-2}$) and 0.029 col. m$^{-2}$ ± 0.011 and 0.012 col. m$^{-2}$ ± 0.005 (range 0–0.16 col. m$^{-2}$) for *V. mirabilis*, respectively, in D9 and D10. Several echinoderms were also observed, including the echinoid Cidaridae (Linnaeus, 1758), the crinoid *Leptometra phalangium* (Müller, 1841), and some holoturians (Figure 5E,F). Overall, the presence of nine species of fish was recorded (Table 2 and Figure 5G–I).

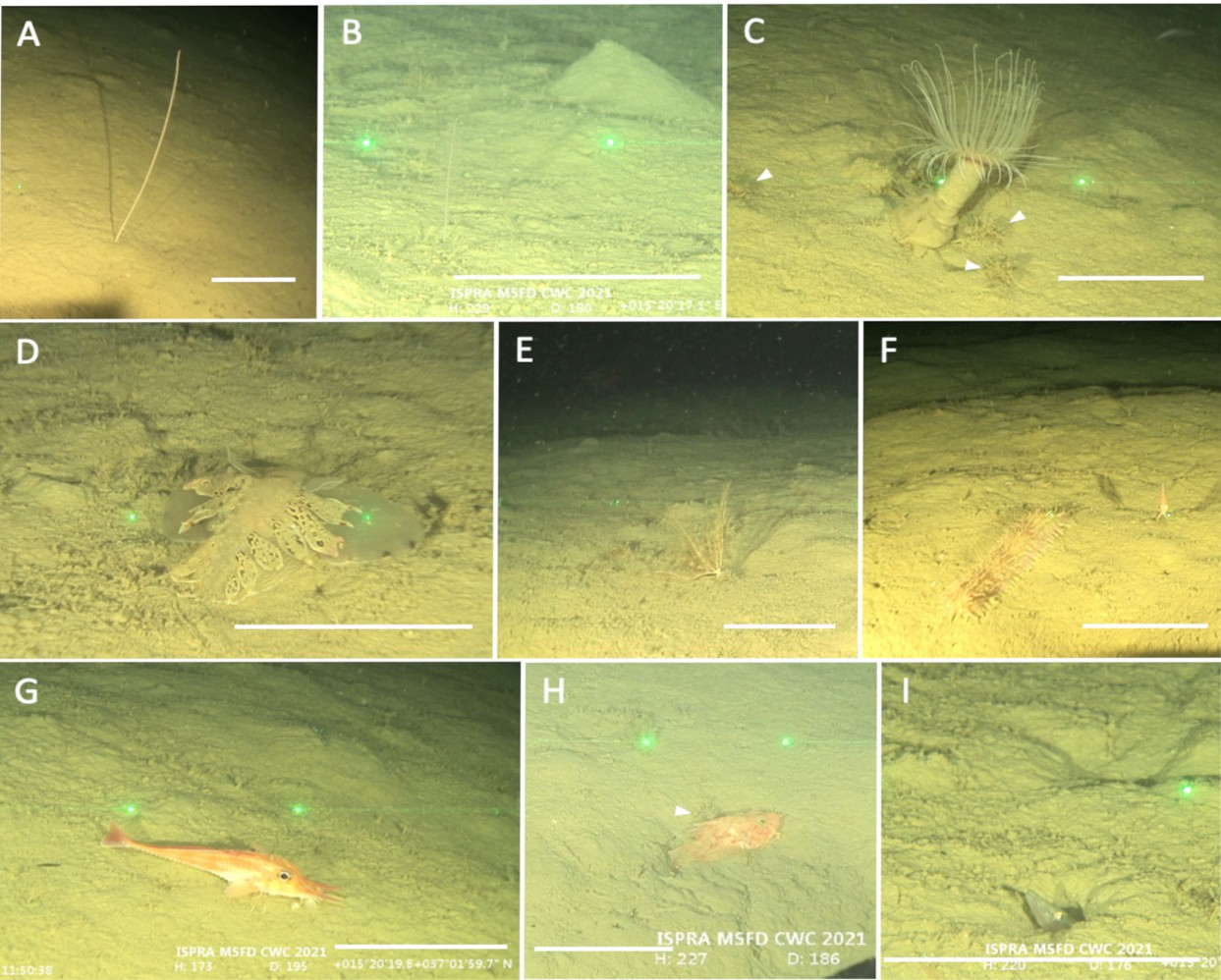

**Figure 5.** Megaebenthic species of the study areas. (**A**) Funiculina quadrangularis; (**B**) Virgularia mirabilis; (**C**) Cerianthus membranaceus and Amage adspersa (white arrows); (**D**) Tethys fimbria; (**E**) Leptometra phalangium; (**F**) Parastichopus regalis and Macroramphosus scolopax; (**G**) Peristedion cataphractum; (**H**) Scorpaena elongata and Amage adspersa (white arrow); (**I**) Ophisurus serpens. Scale bar: 16 cm.

### 3.2.3. Anthropogenic Impact

Macro-litter items were recorded along both tracks, but they were not uniformly distributed, as there were differences in composition and abundance between dives (Table 1). A total of 49 litter items were recorded. The dominant litter types (65.3%) were artificial polymers (Figure 6A): fragments, bags, and bottles made up the most significant portion of the litter (Figure 6B); followed by glass/ceramics (18.4%) and metal items (8.2%). ALDFG mainly consisted of fragments of line, rope, and bricks, accounting for 26.5% of total litter (Figure 7). Although litter of each size class was present at both sites, the most common size class was Class 1 (<1 m²), mainly consisting of plastic fragments, bags, bottles, and plastic glasses. The largest classes were mostly related to larger objects, such as lost hand luggage (Figure 7C). Most of the litter items were observed lying on the seafloor (54.2–68.4% of the total number of observed items). However, some buried items were observed, making it difficult to identify them correctly (8.2% of cases). Hanging items were not found. Of the total litter items, none were observed entangling sessile invertebrates. The presence of epibionts was observed on larger items, with unidentified bryozoans, sponges, and bivalves settled on their surfaces. In some cases, *A. adspersa* tubes were observed in proximity to litter items. The average litter density ranged between 4.78 ± 1.9 items 100 m⁻² and 3.53 ± 1.2 items 100 m⁻², respectively in D9 and D10 (Table 1). Moreover, signs of trawling activities were observed in the area (Figure 7F).

Furthermore, three micro-litter items, consisting of a green plastic fiber (Figure S3) encrusted with organic materials and two thin nylon filaments, were recorded in one of the *A. adspersa* tube samples. The items were embedded in the vegetal fibers of *A. adspersa* but were not used by the worms to build the tube.

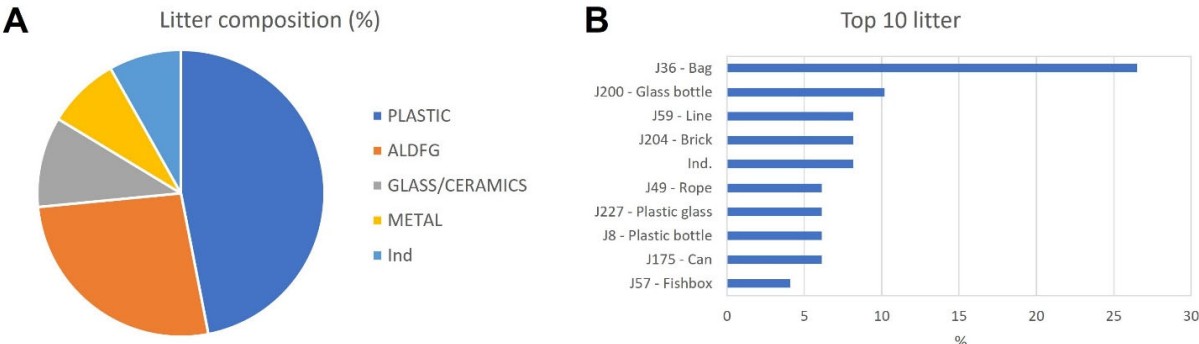

**Figure 6.** Litter composition in the study area. (**A**) Percentage of seafloor macrolitter categories; (**B**) percentage of Top-10 seafloor litter items (%).

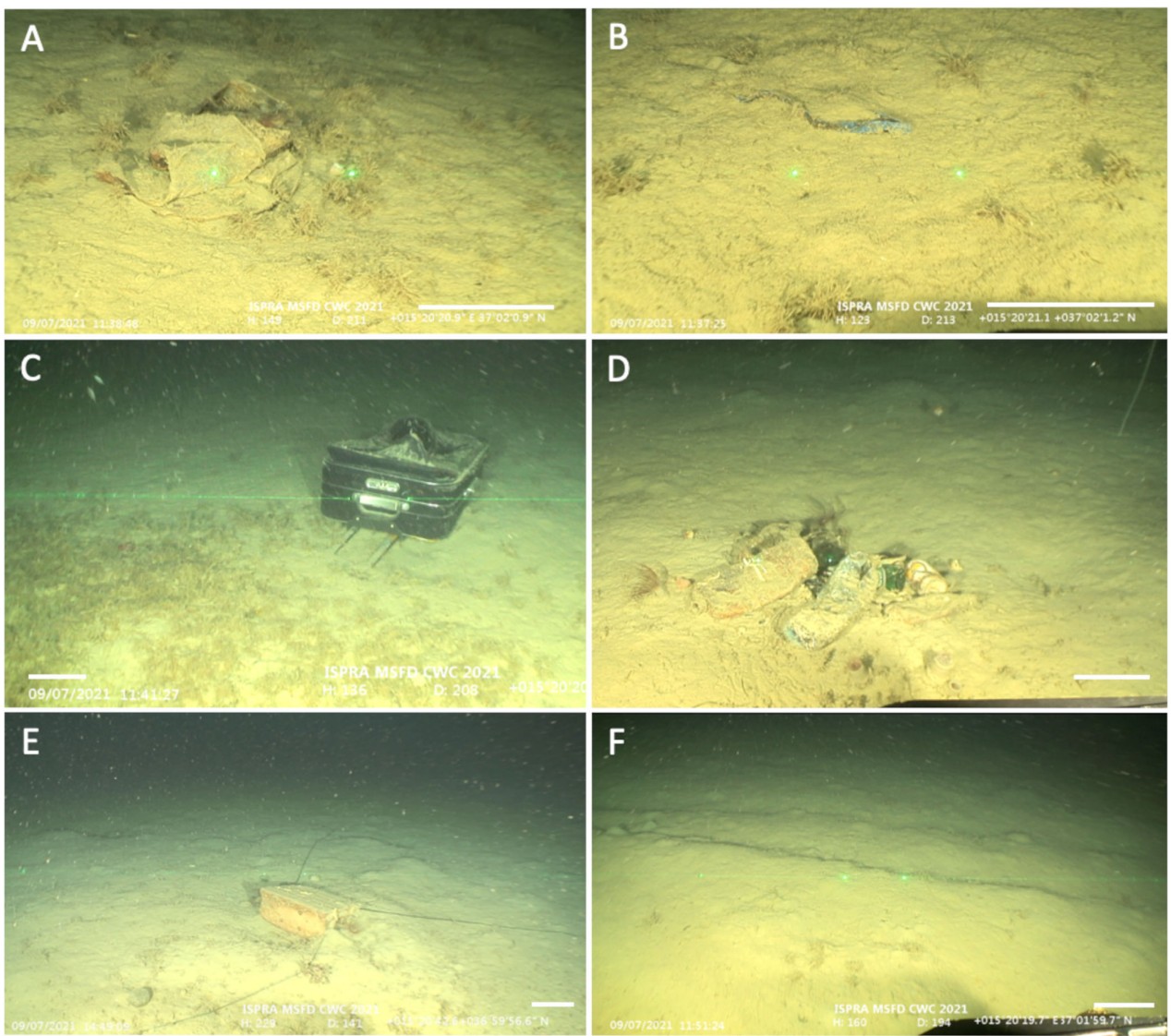

**Figure 7.** Anthropogenic impact in the study areas. (**A**) Plastic bag surrounded by *Amage adspersa*. (**B**) Plastic fragment. (**C**) Lost cabin luggage near dense patches of *A. adspersa*. (**D**) Cans and plastic and glass bottles lying on the seafloor. (**E**) A brick used as shelter by three *Bonellia viridis*. (**F**) Trawl marks. Scale bar: 16 cm.

## 4. Discussion

This study presents the first description of in vivo assemblages of *Amage adspersa* capable of creating dense aggregations in the Ionian Sea (central Mediterranean Sea) on mesophotic and deeper soft seafloor (166–236 m). Data regarding or observations of the ecology of this species are rather scarce. The original description of *A. adspersa* from Lussin (Croatia, northeastern Adriatic Sea) [62] did not provide information about the sediment, depth, and habitat where type material was collected. In the Gulf of Naples, specimens of this species were found routinely in the detrital bottom at a depth of 35 m, where the tube merges with the detrital mass that coats the bottom, more rarely in the mud mixed with sand, and among coralline algae at 65 m depth [44]. In the Tyrrhenian Sea, *A. adspersa* is reported on muddy–detritic bottoms in the bathymetric range of 40 to 90 m [63,64]. The species was found at a depth of 40 to 70 m in the Gulf of Lion in coralline algal bottom [37,65] and at a 60 m depth in Egypt on enteropneust sediments [66]. The deepest recorded

specimens were found at a 114 m depth in the North Sea, while a single record of a sighting at a 500 m depth in the Aegean Sea seems to be unreliable [36]. Thus, to our knowledge, the records from this study are the deepest found in the Mediterranean Sea for this species. Most of the cited records report the presence of few specimens or very low densities, except for [67] who found the species abundant on circalittoral sediments at a 112 m depth, in a location (Ognina Bay, Sicily) about 30 nm from our surveyed sites. In this study, the density of *A. adspersa* tubes reached a maximum of 297.2 tubes m$^{-2}$. During the survey, it was not possible to perform a direct representative sampling of the tubes in the studied area. Nonetheless, out of two sampled tubes, both harbored live specimens of *A. adspersa*. Based on this limited sampling, it can be assumed that a significant proportion of the tubes in the area were likely to be inhabited by living specimens. If confirmed, this would be the highest density found for this species.

Ampharetidae are deposit feeders and show the greatest densities in areas where food and resources naturally accumulate, such as embayments or inlets. Exceptional values of abundance were reported particularly in the deeper dive D9, closer to coastline and urban centers. Some authors have pointed out that an unusual abundance of annelids could be linked to high organic input created by anthropogenic activities [68,69]. The proximity of the study area (particularly dive D9) to urbanized coasts and large harbors (i.e., Siracusa, Augusta, and Catania) could influence species distribution and abundance. The presence of filter-feeders, such as large sea pens, could also support this assumption. However, the absence of detailed data on small-scale oceanographic circulation and on environmental condition prevented us from thoroughly discussing these results, and caution should be exercised.

The availability of *Posidonia oceanica* debris appears to play a key role in the tube-building process by *A. adspersa* [37,38,44,66], at least for some Mediterranean populations. No information is available about the tube features of *A. adspersa* populations outside the Mediterranean Basin [40–43]. [44] found the species occurring frequently on seafloor patches with phanerogams detrital aggregates overlying the sediment surface. On the other hand, none of the recent studies carried out on the macrofaunal communities associated with the *P. oceanica* accumulations of exported macrophytodetritus have reported the presence of *A. adspersa* in similar habitats [70–73]. It is possible that *A. adspersa* may be unable to settle on shallower areas covered with *P. oceanica* macrodetritus, probably because of its surface deposit feeding mode, which may be affected by the presence of massive vegetal debris.

In our study area, there was no evidence of patches of *P. oceanica* macrodetritus on the seabed. Nonetheless, extensive *P. oceanica* meadows are present along the coast at about 1 km from the investigated site (Figure 1), representing a probable source of fine sinking vegetal debris. Since *P. oceanica* meadows export 10–55% of their net primary production as dead matter to nearby habitats [74], *A. adspersa* probably uses the small fibers that fall from the decomposition of the macrodetritus in the upper levels to build its tube. This finding reveals how some marine animals can utilize the dead tissues of *P. oceanica* to construct shelter, highlighting interactions between polychaetes and dead matter from seagrasses in the marine environment and their ecological role.

At the investigated site, the tubes of *A. adspersa* are temporary structures on the surface sediment, covering up to 23.7% of the surveyed seafloor. Dense tube aggregations of ampharetids are known to alter the characteristics of the surrounding sediments and influence the food web on the seabed by contributing and providing habitat and food sources for other marine organisms [25,75]. Indicators of the presence of small organisms within the fibers, such as the mucous tubes of small polychaetes and foraminifers (Figures S3–S5), seems to suggest that the tubes' aggregation of *A. adspersa* may provide microhabitats for smaller organisms of the size of micro- and meiofauna [26]. The presence of these dense tube aggregations suggests that *A. adspersa* likely plays a structuring/habitat-forming role in mesophotic and deeper soft seafloor. However, the ecological role and function of these dense aggregations created by this species remain unclear and require further

investigations. It is important to examine how these aggregations relate to different environmental conditions in order to better understand their significance in ecosystem function.

In the study area, 27 megabenthic species were reported, with fish, mollusks, and echinoderms predominating. From a conservation perspective, the presence of the tall sea pen species *F. quadrangularis* is relevant. It is listed as "vulnerable" in the International Union for Conservation of Nature (IUCN) Red List of Threatened Species both in the Mediterranean Sea and in the Italian checklist [76,77]. Additionally, it has been included in the list of indicator taxa of Vulnerable Marine Ecosystems (VMEs) by the General Fisheries Commission for the Mediterranean Sea [78]. This species is primarily threatened by the direct and indirect effects of trawling [79–81], and this activity is also present in proximity to the study area, as documented by Global Fishing Watch [82] and Marine Traffic [83], as well as through in vivo observations (Figure 7F). The density values of *F. quadrangularis* are comparable to or lower than those reported in literature: i.e., 0.83 colony m$^{-2}$ in the Adriatic Sea at a 162 m depth [79] and from 54.7 to 7771.6 ind. km$^{-2}$ in the Northern and Central Adriatic Sea [81]; along the Gioia Canyon in the southern Tyrrhenian Sea [80], the reported mean density is 0.05–0.35 colony m$^{-2}$.

Sea pens have experienced a decline in the Mediterranean since the 1970s due to intensive trawling activities [84–86]. This group of Anthozoa plays a key role in homogenous muddy environments, supporting biodiversity and providing ecosystem services [86,87]. Thus, this study adds important new data on the presence and/or abundance of sea pen species in central and deep Mediterranean areas, contributing to increasing knowledge of the distribution of these Mediterranean VMEs and providing an assessment of conservation status [85,86].

Another important anthropogenic impact on the benthic community of the investigated area was represented by the huge amount of litter. Unlike the type of litter typically found on rocky features, most of the litter in this study consisted of artificial polymers not typically linked to fishing activities. It was mainly composed of small urban solid waste items such as plastic bags and bottles, glass items, and other plastic fragments. Due to the poor preservation and partial burial status of some litter items, their origin and composition could not be precisely identified. The abundance of domestic items suggests that most of them may originate from land-based sources, although a marine-based source cannot be completely excluded. Some dense aggregations of *A. adspersa* were also found close to litter items, such as some aggregation of *B. viridis* using bricks as a shelter. Similar behavior has already been observed by several authors [88–91], especially in soft sediments where there is a lack of natural structures that function as refuge/shelters. Furthermore, the presence of microplastics observed in a tube of *A. adspesa* could be related to the degradation of macroplastic items [92]. ROV footage and sample analysis suggest how the presence of marine litter is altering the natural environment and community structure, causing changes in ecosystem functioning with still-unclear consequences [90,93,94].

ROV exploration in the Ionian Sea has allowed us to provide an overview of the status of mesophotic and deeper soft seafloor communities of the study area. A great deal of data has been collected, providing new insights into the distribution patterns of important structuring species, thereby enhancing knowledge of VMEs and the impact of marine litter in deep environments. Furthermore, the newly collected data have unveiled previously unknown information about the undisclosed distribution and ecology of *A. adspersa* in the deep waters of Mediterranean Sea. These findings highlight the significance of Amphareti­dae species in increasing the spatial complexity of soft seafloor. However, further data are needed to understand the ecological role played by these soft-seafloor structuring species and to expand knowledge at different spatial scales.

**Supplementary Materials:** The following supporting information can be downloaded at: https://www.mdpi.com/article/10.3390/d15080906/s1, Figure S1: Holotype borrowed from the collections of Museum für Naturkunde (MFN) in Berlin for species comparison; Figure S2: Detail of

the initial section of the tube of *Amage adspersa*; Figure S3: Detail of the terminal opening of the tube of *Amage adspersa*; Figure S4: Detail of the tube of *Amage adspersa*; Figure S5: Remains of organisms attached to the fibers of *Posidonia oceanica* in the distal part of the *Amage adspersa* tube.

**Author Contributions:** Conceptualization and methodology, M.A., F.B. and L.G.; investigation, M.A., M.G. and S.F.R.; identification of biological samples, F.B., L.G. and D.V.; video processing, M.A.; data curation, M.A., F.B., L.G. and M.L.; writing—original draft preparation, M.A., F.B. and L.G.; review and editing, all; funding acquisition, L.T. All authors have read and agreed to the published version of the manuscript.

**Funding:** This research was funded by the Italian Ministry of the Italian Environmental and Energy Security within the framework of the "Marine Strategy Framework Monitoring Program" (PR IS-PRA X0SM0001, 2018).

**Data Availability Statement:** The data presented in this study are available on request from the corresponding author.

**Acknowledgments:** We would like to thank the crew of *R/V Astrea* of *ISPRA* and our colleagues Lorenzo Rossi, Alfredo Pazzini, and Alessia Izzi of ISPRA Marine Robotic Department (Sezione per lo sviluppo tecnologico e supporto del monitoraggio e della ricerca applicata all'ambiente marino profondo) for their professional collaboration and helpfulness in the collection of data. Thanks to Birger Neuhaus, Scientific Head Collection "Vermes" of Museum für Naturkunde, Berlin for providing the holotype of *Amage adspersa*.

**Conflicts of Interest:** The authors declare no conflict of interest.

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
