# Peer review of "Deep Aggregations of the Polychaete Amage adspersa (Grube, 1863) in the Ionian Sea (Central Mediterranean Sea) as Revealed via ROV Observations"

_diversity, doi:10.3390/d15080906_

Round 1
Reviewer 1 Report
The manuscript "Deep aggregations of the polychaete Amage adspersa (Grube, 1863) in the Ionian Sea (Central Mediterranean Sea) as revealed by ROV observations" is a new and interesting contribution to the biology and ecology of polychaetes. It describes a new case of aggregations of a polychaete species. Within annelids or polychaetes this species so far is unknown to occur in dense aggregations. The authors state that such species may act as ecosystem engineers which is suggested also for the current example. However, so far no evidence is presented that this i salso the case for A. adspersa. The observations and as such the publication of these are at this state preliminary because any experimental effort is missing to prove whether this species really influences the ecosystem. For instance, in case of Arenicola or Sabellaria it is known that these aggregations clearly affect the ecosystem such as species composition outside the focus species. Such observations are so far lacking but should be undertaken as soon as possible. The only hint I found was: "Indicators of presence of small organisms within the fibers, such as mucous tubes of small polychaetes and foraminifers, seems to suggest that tubes aggregation of A. adspersa may provide microhabitats for smaller (lines 415-417) but this statement seems not to be validated by data.
organisms of the size of micro- and meiofauna.
On page 14 the authors inform us about interesting results on sea pens and occurrence of litter probably originating from land sources. Both facts are not represented in the title and in the abstract they are not prominently mentioned. Especially the litter problem deserves a more prominentappearance in order not to become lost to the community.
A minor point:
In line 91 the authros speak of the "deep sea" but as I got it right their observations are not deep sea observations but on th econtinental shelf aren't they?
no coments here
Author Response
Response to Reviewers
Manuscript ID: diversity-2509790
Deep aggregations of the polychaete Amage adspersa (Grube, 1863) in the Ionian Sea (Central Mediterranean Sea) as revealed by ROV observations
Dear editor,
We really appreciate the comments made by the two external referees, as well as by the Editor of Diversity. We have taken in consideration all these comments, which have served us to improve the manuscript, and now we hope it will fit with Diversity Journal norms and aims.
A point to point reply to every comment made by the referees is below; in red color you will find our answer/explanation and how it has been implemented in the new manuscript. The revisions made to the manuscript are in track change.
Thank you in advance and please receive our best regards.
Michela Angiolillo and co-authors
Reviewer 1
The manuscript "Deep aggregations of the polychaete Amage adspersa (Grube, 1863) in the Ionian Sea (Central Mediterranean Sea) as revealed by ROV observations" is a new and interesting contribution to the biology and ecology of polychaetes. It describes a new case of aggregations of a polychaete species. Within annelids or polychaetes this species so far is unknown to occur in dense aggregations. The authors state that such species may act as ecosystem engineers which is suggested also for the current example. However, so far no evidence is presented that this is also the case for A. adspersa. The observations and as such the publication of these are at this state preliminary because any experimental effort is missing to prove whether this species really influences the ecosystem. For instance, in case of Arenicola or Sabellaria it is known that these aggregations clearly affect the ecosystem such as species composition outside the focus species. Such observations are so far lacking but should be undertaken as soon as possible. The only hint I found was: "Indicators of presence of small organisms within the fibers, such as mucous tubes of small polychaetes and foraminifers, seems to suggest that tubes aggregation of A. adspersa may provide microhabitats for smaller organisms of the size of micro- and meiofauna (lines 415-417) but this statement seems not to be validated by data.
Response: Thanks for your comments. In the supplementary materials are provided some photos of the organisms dwelling within Amage fibers. In the discussion, we have added the reference to that figures, to help the readers to validate the data. Of course, these are preliminary observations which allowed us to formulate some hypotheses that will need a dedicated experimental sampling design to be tested.
On page 14 the authors inform us about interesting results on sea pens and occurrence of litter probably originating from land sources. Both facts are not represented in the title and in the abstract they are not prominently mentioned. Especially the litter problem deserves a more prominent appearance in order not to become lost to the community.
Response: Thank you for the comments. In order to give more importance to the presence of sea pens and occurrence of litter, in the abstract we have added the respective abundance values. We feel that the title is already long enough, so we have not changed it. Instead, we have added “marine litter” and “VMEs” as keywords
A minor point:
In line 91 the authros speak of the "deep sea" but as I got it right their observations are not deep sea observations but on th econtinental shelf aren't they?
The deep sea is a broad definition that may indicate the ocean depth where light begins to fade (approximately 200 metres depth) or the point of transition from continental shelves to continental slopes. The sentence to which Reviewer 1 is referring to a citation form Wolff (1976) which describes the utilization of seagrass debris by organisms in waters below the photic zone, where decomposition processes only can occur. Therefore, to avoid confusion we choose to substitute “deep sea” with a most generic terminology (i.e. deeper waters).
Reviewer 2 Report
The manuscript by Angiolillo and co-Authors is an interesting work detailing the occurrence of assemblages dominated by the ampharetid Amage adspersa and discussing its possible role as structuring species on mesophotic soft bottoms. I do not have any major comment on the structure of the manuscript, which is mostly satisfying, and I commend the effort made by the Authors to examine the type material, even if I would have included some pictures of the types as well and I recommend to include them if it is possible. The only missing part would have been including some molecular data (e.g., COI sequences), but it is another minor point.
I found just some minor points that deserve some attention before the manuscript can be accepted.
Line 41. “The most referred organisms”: this expression is not very clear; do the Authors mean “the most studied bio-constructing organisms” or “the best-known case”?
Line 77: I am actually convinced that extremely wide geographical and bathymetric distributions are mostly due to mistakes surrounding their identity and/or incomplete data. Usually, when a species is known from an extremely wide range, it is a species complex, and Ampharetidae have been the object of only few studies taking into account also molecular data.
Line 88: Aside from the typo (“matter” instead of “matte”), since this is originally a French term, I would consider putting it in italics. An other way to put it would be “root mat”, but Posidonia matte mostly consists of rhizomes, so “root mat” would be inaccurate.
Lines 122-125: Please state the depth (even approximate, as I expect some seasonal/yearly variation) of the intermediate and deep water layers.
Line 132: The sub-section (2.2. Study area) is clearly wrong, as the previous sub-section (2.1.) has the same title.
Line 158: Even though VLC is a free work environment for the treatment of multimedia files (rather than a software), I think that a citation of the developing team is needed. The same accounts for ImageJ (line 168).
Line 176: I do not understand what the Authors mean by “evaluated […] by occurrence (%)”, as it is not clear to what refers the percentage value (percentage on material? Percentage on use?). I recommend to clarify this point.
Lines 247-249: Add references backing the known distribution of A. adspersa.
Line 299: As much as I know, a reliable discrimination between Cidaris cidaris and Stylocidaris affinis based on ROV footage is not possible; did the Authors sample some material to back their identification?
Table 2: Please include in the table authority and year of the description for all taxa identified to the species level.
Figure 7: The caption refers to “litter”; however, the Authors included trawl marks as well. I suggest to rename the figure as “Anthropic impacts” or something to this effect.
Line 434: As far as I know, according to MDPI formatting rules, web sites should be included among the references.
Further, minor corrections (mostly regarding the language) are included in a modified version of the manuscript. I recommend an additional check of the English expression, because English is not my first language and I could have missed some mistakes.

The English is overall good and understandable, with just a few typos and small grammar errors. I included some corrections in a modified version of the manuscript (see above).
Author Response
Response to Reviewers
Manuscript ID: diversity-2509790
Deep aggregations of the polychaete Amage adspersa (Grube, 1863) in the Ionian Sea (Central Mediterranean Sea) as revealed by ROV observations
Dear editor,
We really appreciate the comments made by the two external referees, as well as by the Editor of Diversity. We have taken in consideration all these comments, which have served us to improve the manuscript, and now we hope it will fit with Diversity Journal norms and aims.
A point to point reply to every comment made by the referees is below; in red color you will find our answer/explanation and how it has been implemented in the new manuscript. The revisions made to the manuscript are in track change.
Thank you in advance and please receive our best regards.
Michela Angiolillo and co-authors
Reviewer 2
The manuscript by Angiolillo and co-Authors is an interesting work detailing the occurrence of assemblages dominated by the ampharetid Amage adspersa and discussing its possible role as structuring species on mesophotic soft bottoms. I do not have any major comment on the structure of the manuscript, which is mostly satisfying, and I commend the effort made by the Authors to examine the type material, even if I would have included some pictures of the types as well and I recommend to include them if it is possible. The only missing part would have been including some molecular data (e.g., COI sequences), but it is another minor point.
Response: Thank you for the comment. As suggested, we have included in the Supplementary material a picture of the holotype. An attempt to do genetic analysis was made, but no consistent results was obtained. As a consequence, this part was not included in the manuscript.
I found just some minor points that deserve some attention before the manuscript can be accepted.
Line 41. “The most referred organisms”: this expression is not very clear; do the Authors mean “the most studied bio-constructing organisms” or “the best-known case”?
Response: Thank you for the suggestion. We have changed the text accordingly.
Line 77: I am actually convinced that extremely wide geographical and bathymetric distributions are mostly due to mistakes surrounding their identity and/or incomplete data. Usually, when a species is known from an extremely wide range, it is a species complex, and Ampharetidae have been the object of only few studies taking into account also molecular data.
Response: Thank you for your comment. The Authors also suspected that despite the morphological similarities there might be significant differences from a molecular point of view between populations from different depths. However, the paucity of available material and some amplification problems due to the primers used did not allow us to test this hypothesis. This will require further sampling aimed at clarifying these aspects.
Line 88: Aside from the typo (“matter” instead of “matte”), since this is originally a French term, I would consider putting it in italics. An other way to put it would be “root mat”, but Posidonia matte mostly consists of rhizomes, so “root mat” would be inaccurate.
Response: We didn’t refer to the matte of Posidonia, but to dead organic matter of seagrass. We reworded adding “organic”.
Lines 122-125: Please state the depth (even approximate, as I expect some seasonal/yearly variation) of the intermediate and deep water layers
Response: Thank you. As suggested, we add the approximate depth ranges of the intermediate and deep water layers.
Line 132: The sub-section (2.2. Study area) is clearly wrong, as the previous sub-section (2.1.) has the same title.
Response: Thank you. It is a typo.
Line 158: Even though VLC is a free work environment for the treatment of multimedia files (rather than a software), I think that a citation of the developing team is needed. The same accounts for ImageJ (line 168).
Response: Done.
Line 176: I do not understand what the Authors mean by “evaluated […] by occurrence (%)”, as it is not clear to what refers the percentage value (percentage on material? Percentage on use?). I recommend to clarify this point.
Response: It refers to the percentage of occurrence in the SU. We have reworded the sentence in order to enhance the clarity.
Lines 247-249: Add references backing the known distribution of A. adspersa.
Response: Done.
Line 299: As much as I know, a reliable discrimination between Cidaris cidaris and Stylocidaris affinis based on ROV footage is not possible; did the Authors sample some material to back their identification?
Response: No, there is no sample for the identification. So, we have corrected in Cidaridae.
Table 2: Please include in the table authority and year of the description for all taxa identified to the species level.
Response: Done
Figure 7: The caption refers to “litter”; however, the Authors included trawl marks as well. I suggest to rename the figure as “Anthropic impacts” or something to this effect.
Response: Thank you. We have changed the text accordingly.
Line 434: As far as I know, according to MDPI formatting rules, web sites should be included among the references.
Response: Done. We have included the quoted web sites among the references.
Further, minor corrections (mostly regarding the language) are included in a modified version of the manuscript. I recommend an additional check of the English expression, because English is not my first language and I could have missed some mistakes.
Response: Thank you, we added all the correction in the revised version of the manuscript.
Comments on the Quality of English Language
The English is overall good and understandable, with just a few typos and small grammar errors. I included some corrections in a modified version of the manuscript (see above).
Response: Thank you for the corrections. We have included them in the revised version of the manuscript.